**Data Availability Statement:** No datasets were generated or analysed during the current study. All

# Trends in using intraoperative parathyroid hormone monitoring during parathyroidectomy: Protocol and rationale for a cross-sectional survey study of North American surgeons

**Phillip Staibano**[1,2]*, **Tyler McKechnie**[2,3], **Alex Thabane**[2], **Michael Xie**[1], **Han Zhang**[1], **Michael K. Gupta**[1], **Michael Au**[1], **Jesse D. Pasternak**[4], **Sameer Parpia**[2], **James Edward Massey Young**[1], **Mohit Bhandari**[2,5]

1 Division of Otolaryngology–Head and Neck Surgery, Department of Surgery, McMaster University, Hamilton, Ontario, Canada, 2 Department of Health Research Methods, Evidence, and Impact, McMaster University, Hamilton, ON, Canada, 3 Division of General Surgery, Department of Surgery, McMaster University, Hamilton, Ontario, Canada, 4 Endocrine Surgery Section Head, Division of General Surgery, Department of Surgery, University Health Network, University of Toronto, Toronto, Ontario, Canada, 5 Division of Orthopedic Surgery, Department of Surgery, McMaster University, Hamilton, Ontario, Canada

* phillip.staibano@medportal.ca

## Abstract

Hyperparathyroidism is a common endocrine disorder that occurs secondary to abnormal parathyroid gland functioning. Depending on the type of hyperparathyroidism, surgical extirpation of hyperfunctioning parathyroid glands can be considered for disease cure. Intraoperative parathyroid hormone (IOPTH) monitoring improves outcomes in patients undergoing surgery for primary hyperparathyroidism, but studies are needed to characterize its institutional adoption and its role in surgery for secondary and tertiary hyperparathyroidism, as these entities can be difficult to cure. Hence, we will perform a cross-sectional survey study of surgeon rationale, operational details, and barriers associated with IOPTH monitoring adoption across North America. We will utilize a convenience sampling technique to distribute an online survey to head and neck surgeons and endocrine surgeons across North America. This survey will be distributed via email to three North American professional societies (i.e., Canadian Society for Otolaryngologists–Head and Neck Surgeons, American Head and Neck Society, and American Association of Endocrine Surgeons). The survey will consist of 30 multiple choice questions that are divided into three concepts: (1) participant demographics and training details, (2) details of surgical adjuncts during parathyroidectomy, and (3) barriers to adoption of IOPTH. Descriptive analyses and multiple logistic regression will be used to evaluate the impact of demographic, institutional, and training variables on the use of IOPTH monitoring in surgery for all types of hyperparathyroidism and barriers to IOPTH monitoring adoption. Ethics approval was obtained by the Hamilton Integrated Research Ethics Board (2024-17173-GRA). These findings will characterize surgeon and institutional practices with regards to IOPTH monitoring during parathyroid surgery and

relevant data from this study will be made available upon study completion.

**Funding:** The author(s) received no specific funding for this work.

will inform future trials aimed to optimize the use of IOPTH monitoring in secondary and tertiary hyperparathyroidism.

## Introduction

Hyperparathyroidism is defined by abnormal secretion of parathyroid hormone [1]. Primary hyperparathyroidism is caused by a single hyperfunctioning gland in over 80% of cases and it is classically associated with osteoporosis, nephrocalcinosis, and nephrolithiasis [2]. The global incidence of primary hyperparathyroidism is expected to increase alongside wider accessibility to serum calcium measurements and it remains one of the most common endocrine disorders amongst postmenopausal women [3]. Secondary and tertiary hyperparathyroidism, on the other hand, are typically diagnosed in the context of chronic kidney disease, which can lead to persistent parathyroid gland stimulation and hyperplastic changes affecting more than one parathyroid gland [4]. Secondary hyperparathyroidism is associated with worse cardiovascular and mortality outcomes in CKD and will likely rise in prevalence alongside increasing rates of CKD while tertiary hyperparathyroidism is becoming more prevalent secondary to increased global access to hemodialysis and kidney transplantation [5, 6]. In primary hyperparathyroidism, surgical extirpation of hyperfunctioning gland tissue is often indicated for disease cure [7]. In 2017, clinical practice guidelines for chronic kidney disease-mineral and bone disorder secondary recommended parathyroidectomy as preferred therapy for refractory secondary hyperparathyroidism [8]. While in tertiary hyperparathyroidism, despite advances in medical therapy with calcimimetics, parathyroidectomy represents definitive management with higher cure rates and lower complication rates compared to chronic medical therapy [9, 10]. Parathyroidectomy is a commonly performed surgery within the technical scope of head and neck surgeons and endocrine surgeons [1, 11]. Despite advances in imaging that have improved cure rates in primary hyperparathyroidism, the surgical management of secondary and tertiary hyperparathyroidism remains challenging [12].

In 1991, Irvin and colleagues successfully harnessed the rapid half-life of PTH to guide parathyroidectomy by introducing intraoperative PTH (IOPTH) monitoring [13]. Since then, this technology has been used increasingly in parathyroidectomy with recent meta-analyses demonstrating its utility in reducing persistent and recurrent disease following surgery [14, 15]. The benefit, however, of IOPTH monitoring in surgery for secondary and tertiary hyperparathyroidism remains controversial [16]. Herein, we report a protocol for cross-sectional survey study of IOPTH monitoring practices amongst head and neck and endocrine surgeons across North America. The aims of this study are twofold: (1) To describe the clinical indications and technical details regarding IOPTH monitoring in primary, secondary, and tertiary hyperparathyroidism; (2) To identify the clinical, geographic, and institutional barriers to adopting IOPTH monitoring at endocrine centres across North America.

## Materials and methods

### Sampling technique and recruitment

We will perform convenience sampling to recruit survey respondents for this descriptive survey study. All candidate respondents will be head and neck surgeons and endocrine surgeons practicing in North America and they will be contacted via the email lists of professional and academic societies (e.g., American Head and neck Society, American Association of Endocrine Surgeons, and Canadian Society of Otolaryngologists–Head and Neck Surgeons). It is important to note that active AAES members may reside in North, Central, and South America [17].

As these professional societies may not differentiate staff surgeons from surgical trainees, we will evaluate this based on survey responses. Our purpose is to sample academic surgeons who work at high-volume endocrine centers and so, distributing this survey via professional society email networks will capture that intended population.

### Description of survey tool and distribution

The online survey will contain a description of the study objectives, benefits, and harms of participating in the study, requirements for participation, the implications of the research findings, and a consent form. The survey will be generated using Qualtrics XM software (Provo, UT, USA) and administered to clinician emails via approved professional society mailing lists. There will be 30 questions requiring multiple choice responses with and without freeform text input. The survey will be divided into three sections: (1) demographics and surgical training, (2) the use of IOPTH and surgical adjuncts during parathyroidectomy, and (3) perceived barriers to adopting IOPTH monitoring (S1 Appendix). There will be adaptive reasoning within the survey based upon respondents' answers to certain questions. Neither patients nor the public were involved in the design of the survey. The survey was created with input from five content experts and was piloted for flow and clinical sensibility with five local surgeons, whereby further refinements were implemented [18]. This survey study will be conducted and reported in accordance with the Checklist for Reporting Results of Internet E-Surveys (CHERRIES) [19].

Survey respondents will not be required to answer every survey question to complete and submit the survey. We estimate it will take 5–8 minutes to complete the entire survey. There will be no non-English translations of the survey as it will only be distributed within North America. All respondents will only be able to submit survey once. Survey completion will not be timed and there will be no incentives for completing the survey. We will not stratify survey distribution based upon demographic variables or training level. The survey will be distributed on a rolling basis over the course of four months (i.e., July 1, 2024, to November 1, 2024) and will be emailed to potential respondents at least twice during this interval. All survey distribution will be overseen by participating professional societies and the study team will not have access to respondent emails. The anticipating study completion date is December 31, 2024. Respondents will not receive their completed survey results but will be notified of publication through their respective professional society.

### Planned sample size

In Canada, there are an estimated 715 otolaryngology–head and neck surgeons practicing in either community or academic centres [20]. In the US, there are an estimated 12,609 otolaryngology–head and neck surgeons practicing in either community or academic centres [21]. There are 518 active or corresponding members in the AAES [17]. We do understand that not all practicing surgeons are members of the professional societies selected for survey distribution and thus, estimate that we will be sampling from a population of approximately 10,000 practicing surgeons and trainees across North America. As the goal of this survey is to describe IOPTH usage trends without a predefined primary outcome, we will not perform a sample size calculation. We will, however, aim for a survey completion rate of 60–75%, which is consistent with other survey studies of surgeons and will help maintain external validity [18, 22].

### Study outcomes

The overall goal of this survey study is to explore clinical indications and technical details for employing IOPTH monitoring in primary, secondary, and tertiary hyperparathyroidism. We

plan on characterizing IOPTH usage trends in the context of type of surgical training, type of surgical practice, and geography of surgical practice. The respondent demographic questions will be explored using nominal variables, while indications and details regarding IOPTH monitoring will be explored using both nominal and ordinal variables (i.e., Likert scale responses).

## Data analysis plan

For all completed or partially completed surveys, we will perform descriptive analyses to report proportions and frequencies for each survey response. We will evaluate variability in responses and estimates using 95% confidence intervals. As all responses will be described as nominal or ordinal variables, we will perform chi-square or Fischer's exact tests for comparative analysis with an emphasis on comparing type of surgical discipline and location of practice. Based on survey responses, we will also evaluate the relationship of surgical practice location and the year of acquisition on the uptake of IOPTH monitoring technology. We also plan to perform logistic regression analysis to determine the effect of demographic and training variables on responses pertaining to rationale and details for using IOPTH monitoring and barriers to using IOPTH monitoring technology. The key characteristics that we will insert in our multiple logistic regression model will be age, gender, type/level of training, type of practice, surgical volume, and location of training and practice. We will utilize the stepwise regression analysis to include significant variables into the model. We will evaluate model assumptions and goodness-of-fit by examining residuals and at least 10 observations will be required for each independent variable to be included into the regression model. There will be no planned subgroup analyses. Any free-text responses will be evaluated based upon conceptual analysis. All statistical analyses will be performed using R (version 4.3.1, Vienna, Austria).

## Ethical and safety considerations

Ethics approval was obtained from the Hamilton Integrated Research Ethics Board as project number 2023-17173-GRA. Participation in this study will be voluntary and all participants will provide informed consent before starting the online survey. Participation will unlikely cause any adverse effects or discomfort related to the conduct or outcomes of the survey. There will be no timer associated with survey completion. All data collected will be pseudonymous. Survey datasets will be securely stored in Qualtrics, protected by Single-Sign-On, and in password-protected OneDrive accounts. The data will reside on password-protected, encrypted personal computers accessed via secure networks. Data will be destroyed two years following the completion of the study. All protocol amendments will be submitted to the Hamilton Integrated Research Ethics Board as modifications prior to implementation. Amendments will also be communicated during dissemination to both academic and lay audiences.

## Discussion

In 2016, the American Association of Endocrine Surgeons strongly recommended the use of IOPTH in minimally invasive parathyroidectomy for primary hyperparathyroidism to reduce operative failure [1]. Furthermore, it has been suggested that IOPTH monitoring may be helpful in guiding challenging parathyroid surgery such as in cases of parathyroid hyperplasia and renal hyperparathyroidism [23, 24]. There is, however, little standardization and consensus in the utility of IOPTH monitoring amongst patients undergoing parathyroid surgery. The survey will explore alternative indications for applying IOPTH monitoring to guide parathyroidectomy and describe the technical details of using IOPTH monitoring at institutions across North America. These results will help to inform future prospective trials aimed at optimizing the use of IOPTH monitoring in challenging parathyroid surgeries.

### Dissemination plan

The findings of the survey will be submitted for publication in a peer-reviewed journal aimed at informing surgical practices in head and neck, and endocrine surgery. The primary investigator will present findings at national and international conferences. We will also communicate study findings to relevant stakeholders (i.e., surgeons, universities, hospital administrations, and government officials) to brainstorm other research questions and translate survey results into actionable items. These results will be used to inform future observational studies, cost analyses, and trials aimed at improving the adoption and optimization of IOPTH technology for guiding parathyroidectomy.

## Supporting information

**S1 Appendix. Survey cover letter and questionnaire.**
(DOCX)

## Author Contributions

**Conceptualization:** Phillip Staibano, Tyler McKechnie, Alex Thabane, Michael Xie, Han Zhang, Michael K. Gupta, Michael Au, Jesse D. Pasternak, Sameer Parpia, James Edward Massey Young.

**Methodology:** Phillip Staibano.

**Project administration:** Phillip Staibano.

**Resources:** Phillip Staibano.

**Writing – original draft:** Phillip Staibano, Tyler McKechnie, Alex Thabane, Michael Xie, Han Zhang, Michael K. Gupta, Michael Au, Jesse D. Pasternak, Sameer Parpia, James Edward Massey Young, Mohit Bhandari.

**Writing – review & editing:** Phillip Staibano, Tyler McKechnie, Alex Thabane, Michael Xie, Han Zhang, Michael K. Gupta, Michael Au, Jesse D. Pasternak, Sameer Parpia, James Edward Massey Young, Mohit Bhandari.

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
