## [Decision Letter · Decision Letter 0]

12 Jun 2024

PONE-D-24-07939Trends in using intraoperative parathyroid monitoring during parathyroidectomy: Protocol and rationale for a cross-sectional survey study of North American surgeonsPLOS ONE

Dear Dr. Staibano,

Thank you for submitting your manuscript to PLOS ONE. After careful consideration, we feel that it has merit but does not fully meet PLOS ONE’s publication criteria as it currently stands. Therefore, we invite you to submit a revised version of the manuscript that addresses the points raised during the review process.

Please provide a point-by-point response to reviewers #2 and #3. Unfortunately reviewer #1 assessed the manuscript as a final paper and not as a protocol. Looking forward to your revised manuscript.

We look forward to receiving your revised manuscript.

Kind regards,

Mabel Aoun, MD, MPH

Academic Editor

PLOS ONE

Journal Requirements:

3. Ethics statement appears in the Methods section of the manuscript AND at the end of the manuscript:

Your ethics statement should only appear in the Methods section of your manuscript. If your ethics statement is written in any section besides the Methods, please delete it from any other section. 

Reviewers' comments:

Reviewer's Responses to Questions

**Comments to the Author**

1. Does the manuscript provide a valid rationale for the proposed study, with clearly identified and justified research questions?

Reviewer #1: No

Reviewer #2: Partly

Reviewer #3: Yes

2. Is the protocol technically sound and planned in a manner that will lead to a meaningful outcome and allow testing the stated hypotheses?

Reviewer #1: No

Reviewer #2: Yes

Reviewer #3: Yes

3. Is the methodology feasible and described in sufficient detail to allow the work to be replicable?

Reviewer #1: No

Reviewer #2: Yes

Reviewer #3: Yes

4. Have the authors described where all data underlying the findings will be made available when the study is complete?

Reviewer #1: No

Reviewer #2: Yes

Reviewer #3: No

5. Is the manuscript presented in an intelligible fashion and written in standard English?

Reviewer #1: No

Reviewer #2: Yes

Reviewer #3: Yes

6. Review Comments to the Author

You may also provide optional suggestions and comments to authors that they might find helpful in planning their study.

Reviewer #1: Thank you for giving me the opportunity to review this manuscript.

This manuscript did not include any study outcomes. I could not evaluate the importance of this manuscript without results. Please re-submit after finalizing the study.

Reviewer #2: In this original manuscript by Staibano et al., authors report the protocole of a survey they plan to start on March, 2024 about the use of intra-operative parathyroid hormone (IOPTH) measurement during parathyroidectomy either for primary, secondary, or tertiary hyperparathyroidism. IOPTH is being more and more used for primary hyperparathyroidism (PHPT) but very few is known about its use when treating secondary (SHPT) or tertiary (THPT) hyperparathyroidism. In such, this study will help the community get a better understanding of both its use and barriers. Manuscript is overall well written and protocole is very straightforward. I still have some comments/questions authors may address.

A. MAJOR COMMENTS

1. Within the introduction section, authors suggest that "all [forms of HPT] are definitively managed via surgical extirpation". Later, they suggest that SHPT may be managed medically. So, that first statement may be more tempered: the price for "curing" SHPT may sometimes be hypoparathyroidism. Moreover, more and more data suggest that THPT may recover several months/years after kidney transplantation without surgery.

2. Authors suggest that SHPT and THPT only occur during chronic kidney disease (CKD) which, if probably frequent, is not totally true: SHPT is even more frequent during vitamin D deficiency or obesity. Here, surgery should never be performed. Moreover, authors suggest that THPT may only be diagnosed after kidney transplant which, again, if frequent, is not the sole circumstance for diagnosis, dialysis being probably even more frequent.

B. MINOR COMMENTS

1. In the description of the study, authors mention the survey will be distributed for 6 months. A few sentences later, they state it will take place from March 28 to August 31, meaning 5 months. I'd suggest being consistent between those two statements.

2. Regarding ethics, authors mention their survey will be 'anonymous'. Anonymous data are data that cannot allow identifying people whom data have been collected. In the present study, crossing demographics and educational data may allow to identify participants. Therefore, such dataset is made of pseudonymous data in which no data allow direct identification. I'd strongly suggest to use this wording (pseudonymous) instead of 'anonymous' when describing such a dataset.

3. In the appendix, the letter mentions that data will be stored for 5 years after the completion of the study while the manuscript states it will be for 2 years. I'd suggest being consistent between those two statements.

Reviewer #3: In this paper for a protocol for a survey study of surgeons performing parathyroidectomy is presented. The study design is well presented and the protocol easy to follow.

The authors should specify how they will follow the PLOS Data policy - The PLOS Data policy requires authors to make all data underlying the findings described in their manuscript fully available without restriction, with rare exception, at the time of publication. The data should be provided as part of the manuscript or its supporting information, or deposited to a public repository. For example, in addition to summary statistics, the data points behind means, medians and variance measures should be available. If there are restrictions on publicly sharing data—e.g. participant privacy or use of data from a third party—those must be specified.

7. PLOS authors have the option to publish the peer review history of their article (what does this mean?). If published, this will include your full peer review and any attached files.

Reviewer #1: **Yes: **Takahisa Hiramitsu

Reviewer #2: **Yes: **Jean-Philippe Bertocchio

Reviewer #3: No

---

## [Author Response · Author response to Decision Letter 0]

13 Jun 2024

PONE-D-24-07939

Trends in using intraoperative parathyroid monitoring during parathyroidectomy: Protocol and rationale for a cross-sectional survey study of North American surgeons

PLOS ONE

Dear Dr. Staibano,

Thank you for submitting your manuscript to PLOS ONE. After careful consideration, we feel that it has merit but does not fully meet PLOS ONE’s publication criteria as it currently stands. Therefore, we invite you to submit a revised version of the manuscript that addresses the points raised during the review process.

Please provide a point-by-point response to reviewers #2 and #3. Unfortunately reviewer #1 assessed the manuscript as a final paper and not as a protocol. Looking forward to your revised manuscript.

We look forward to receiving your revised manuscript.

Kind regards,

Mabel Aoun, MD, MPH

Academic Editor

PLOS ONE

Journal Requirements:

Thank you – we did adjust style and file naming to meet PLOS ONE requirements. 

We have made all associated documents available. Since this is a protocol, there is no collected data that requires sharing. All authors agree to full data sharing plan. 

3. Ethics statement appears in the Methods section of the manuscript AND at the end of the manuscript:

Your ethics statement should only appear in the Methods section of your manuscript. If your ethics statement is written in any section besides the Methods, please delete it from any other section.

We have made sure to only include this in the methods section. 

We did change this match formatting guidelines. 

All references have been reviewed. 

Comments to the Author

1. Does the manuscript provide a valid rationale for the proposed study, with clearly identified and justified research questions?

Reviewer #1: No

Reviewer #2: Partly

Reviewer #3: Yes

2. Is the protocol technically sound and planned in a manner that will lead to a meaningful outcome and allow testing the stated hypotheses?

Reviewer #1: No

Reviewer #2: Yes

Reviewer #3: Yes

3. Is the methodology feasible and described in sufficient detail to allow the work to be replicable?

Reviewer #1: No

Reviewer #2: Yes

Reviewer #3: Yes

4. Have the authors described where all data underlying the findings will be made available when the study is complete?

Reviewer #1: No

Reviewer #2: Yes

Reviewer #3: No

5. Is the manuscript presented in an intelligible fashion and written in standard English?

Reviewer #1: No

Reviewer #2: Yes

Reviewer #3: Yes

6. Review Comments to the Author

You may also provide optional suggestions and comments to authors that they might find helpful in planning their study.

Reviewer #1: Thank you for giving me the opportunity to review this manuscript.

This manuscript did not include any study outcomes. I could not evaluate the importance of this manuscript without results. Please re-submit after finalizing the study.

Thank you for taking the time to review this manuscript. 

Reviewer #2: In this original manuscript by Staibano et al., authors report the protocole of a survey they plan to start on March, 2024 about the use of intra-operative parathyroid hormone (IOPTH) measurement during parathyroidectomy either for primary, secondary, or tertiary hyperparathyroidism. IOPTH is being more and more used for primary hyperparathyroidism (PHPT) but very few is known about its use when treating secondary (SHPT) or tertiary (THPT) hyperparathyroidism. In such, this study will help the community get a better understanding of both its use and barriers. Manuscript is overall well written and protocole is very straightforward. I still have some comments/questions authors may address.

 Thank you for taking the time to review our manuscript. 

A. MAJOR COMMENTS

1. Within the introduction section, authors suggest that "all [forms of HPT] are definitively managed via surgical extirpation". Later, they suggest that SHPT may be managed medically. So, that first statement may be more tempered: the price for "curing" SHPT may sometimes be hypoparathyroidism. Moreover, more and more data suggest that THPT may recover several months/years after kidney transplantation without surgery.

Thank you for this comment. In the abstract and introduction sections, we did soften this language. 

2. Authors suggest that SHPT and THPT only occur during chronic kidney disease (CKD) which, if probably frequent, is not totally true: SHPT is even more frequent during vitamin D deficiency or obesity. Here, surgery should never be performed. Moreover, authors suggest that THPT may only be diagnosed after kidney transplant which, again, if frequent, is not the sole circumstance for diagnosis, dialysis being probably even more frequent.

Thank you for this comment. In the introduction section, we did soften this language. 

B. MINOR COMMENTS

1. In the description of the study, authors mention the survey will be distributed for 6 months. A few sentences later, they state it will take place from March 28 to August 31, meaning 5 months. I'd suggest being consistent between those two statements.

To account for delayed processing in the approval of this survey by the professional societies that will oversee distribution, we have updated the time window of survey distribution. 

We have clarified that survey distribution will happen from July 1 to November 1, 2024 and that the study will be completed by December 31, 2024. 

2. Regarding ethics, authors mention their survey will be 'anonymous'. Anonymous data are data that cannot allow identifying people whom data have been collected. In the present study, crossing demographics and educational data may allow to identify participants. Therefore, such dataset is made of pseudonymous data in which no data allow direct identification. I'd strongly suggest to use this wording (pseudonymous) instead of 'anonymous' when describing such a dataset.

Thank you – this is a very helpful comment and we have amended this to pseudonymous. 

3. In the appendix, the letter mentions that data will be stored for 5 years after the completion of the study while the manuscript states it will be for 2 years. I'd suggest being consistent between those two statements.

We did change this to two years on both documents. We also removed the survey link since this leads to a beta version of the survey that we used for validation within our team. The actual link will be sent with the email to potential respondents. 

Reviewer #3: In this paper for a protocol for a survey study of surgeons performing parathyroidectomy is presented. The study design is well presented and the protocol easy to follow.

Thank you for your comments. 

The authors should specify how they will follow the PLOS Data policy - The PLOS Data policy requires authors to make all data underlying the findings described in their manuscript fully available without restriction, with rare exception, at the time of publication. The data should be provided as part of the manuscript or its supporting information, or deposited to a public repository. For example, in addition to summary statistics, the data points behind means, medians and variance measures should be available. If there are restrictions on publicly sharing data—e.g. participant privacy or use of data from a third party—those must be specified.

7. PLOS authors have the option to publish the peer review history of their article (what does this mean?). If published, this will include your full peer review and any attached files.

Do you want your identity to be public for this peer review? For information about this choice, including consent withdrawal, please see our Privacy Policy.

Reviewer #1: Yes: Takahisa Hiramitsu

Reviewer #2: Yes: Jean-Philippe Bertocchio

Reviewer #3: No

---

## [Editor Report · Decision Letter 1]

16 Jun 2024

Trends in using intraoperative parathyroid monitoring hormone during parathyroidectomy: Protocol and rationale for a cross-sectional survey study of North American surgeons

PONE-D-24-07939R1

Dear Dr. Staibano,

We’re pleased to inform you that your manuscript has been judged scientifically suitable for publication and will be formally accepted for publication once it meets all outstanding technical requirements.

Kind regards,

Mabel Aoun, MD, MPH

Academic Editor

PLOS ONE
---

## [Editor Report · Acceptance letter]

27 Jun 2024

PONE-D-24-07939R1 

PLOS ONE

Dear Dr. Staibano, 

I'm pleased to inform you that your manuscript has been deemed suitable for publication in PLOS ONE. Congratulations! Your manuscript is now being handed over to our production team.

Kind regards, 

on behalf of

Dr. Mabel Aoun 

Academic Editor

PLOS ONE